# Environmental risk factors for self-harm during imprisonment: A pilot prospective cohort study

**Thomas Stephenson**[1,2]*, **Imogen Harris**[3], **Charlotte Armstrong**[4], **Seena Fazel**[5,6], **Roxanna Short**[1], **Nigel Blackwood**[1,3]

1 Department of Forensic and Neurodevelopmental Sciences, Institute of Psychiatry Psychology and Neuroscience, King's College London, London, United Kingdom, 2 Oxleas NHS Foundation Trust, London, United Kingdom, 3 South London and Maudsley NHS Foundation Trust, London, United Kingdom, 4 North East London NHS Foundation Trust, London, United Kingdom, 5 Department of Psychiatry, University of Oxford, Oxford, United Kingdom, 6 Oxford Health NHS Foundation Trust, Oxford, United Kingdom

* Thomas.stephenson@kcl.ac.uk

## Abstract

### Introduction

Self-harm is a major public health issue in the imprisoned population. Limited high-quality evidence exists for the potential impact of prison environmental factors such as solitary confinement. This exploratory pilot prospective cohort study in a large male remand prison in England sought to estimate effect sizes for a comprehensive range of prison environmental factors in relation to self-harming behaviours.

### Methods

A random sample of all prisoners (N = 149) starting a period of imprisonment at the study prison took part in a clinical research interview, which assessed a range of known risk factors for self-harm in prison. Information concerning environmental factors, including staff numbers, cell placement and movements, and engagement in work and activities were collected from prison records. Incidents of self-harm behaviour in the 3 months after entering prison were measured using medical records and self-report at end of follow-up. Multivariable logistic regression models were calculated individually for each predictor.

### Results

55.7% of participants completed follow-up (83/149). Single cell placement (OR 4.31, 95% CI 1.06–18.24, p = 0.041) and more frequent changes of cellmate (OR 1.52, CI 1.14–2.17, p = 0.009) and cell (OR 1.83, 95% CI 1.28–2.86, p = 0.003) were associated with an increased risk of self-harming behaviour. Time spent in areas with a higher number of prisoners per member of prison staff was significantly associated with reduced self-harm behaviour in adjusted models (OR 0.89, CI 0.78–0.99, p = 0.039). Following sensitivity analyses, the associations between frequent cell changes and self-harm behaviour, and between single cell placement and self-harm ideation, remained statistically significant.

**Data availability statement:** Data cannot be shared publicly due to concerns that, even if anonymized, participants in this vulnerable population might be identifiable given low counts for many study variables. Requests for study data can be directed to the HM Prison and Probation Service National Research Committee (national.research@justice.gov.uk).

**Funding:** TS received funding for this research under a Preparatory Clinical Research Training Fellowship from the NIHR Maudsley BRC (IS-BRC-1215-20018). SF is funded by the NIHR Oxford Health BRC. The funders had no role in study design, data collection and analysis, decision to publish, or preparation of the manuscript.

**Competing interests:** The authors have declared that no competing interests exist.

## Discussion

This exploratory pilot study provides prospective longitudinal data regarding relationships between prison environmental factors and self-harm behaviour. Findings regarding single cell accommodation and frequent cell changes are consistent with the prior evidence base largely derived from case-control study data. The finding regarding frequent cellmate changes predicting self-harm is novel. Findings regarding prisoner-staff ratio and self-harm most likely reflect a reverse causal relationship. Replication in larger cohort studies is required to address the limitations of this pilot study.

## Introduction

Self-harm is a major public health issue in prisons [1]. Governmental data in England and Wales record the annual incidence of self-harm to have more than doubled in the last decade, from less than 300 per 1,000 prisoners in 2013 to 805 per 1,000 prisoners in September 2023 [2]. Self-harm behaviour in prison is an established risk factor for subsequent suicide, both in prison and on release into the community [1,3].

A recent systematic review and meta-analysis highlighted the empirical evidence for a range of individual characteristics contributing to self-harm behaviour in prison populations, including sociodemographic, criminological, historical, clinical and psychosocial variables [4]. These include self-harm behaviour prior to imprisonment and pre-existing psychiatric morbidity (for example, personality and depressive disorders). High levels of such morbidity have been documented across prison populations in many countries [5]. The same systematic review and meta-analysis also identified moderate-sized effects on self-harm for prison environmental factors such as exposure to solitary confinement, single cell placement, lack of social contact, and lack of employment in prison. One case-control study additionally identified that frequent cell moves within prison over the previous two years was associated with self-harm behaviour [6]. However, there is limited high-quality evidence regarding the longitudinal relationship between events, experiences and environments encountered by prisoners and engagement in self-harm behaviours. A prognostic model developed to help risk assessment in the area does not include any prison environmental variables [7].

Recent work reviewing the available evidence has suggested that prison regime characteristics, namely lower time out of cell and lower time in purposeful activity, may influence self-harm and suicide risk in prisons [8]. In addition, qualitative and cross-sectional research implicates a broader range of potential prison environmental factors for self-harm such as lower staffing levels, difficulties in relationships between prisoners and staff, high staff turnover, high prisoner turnover and overcrowding [9–13]. However, the strength of such potential environmental effects is currently unclear. In addition, most existing studies examine female prisoner populations. Longitudinal study designs could provide high-quality evidence regarding these relationships in both male and female populations, which could in turn inform the clinical assessment and management of self-harm in this population and suicide prevention strategies in prisons, however high turnover rates in prison populations call the feasibility of such approaches into question.

The aims of this exploratory pilot prospective cohort study were therefore to estimate the effect sizes of a range of potentially relevant prison environmental variables in relation to self-harm behaviour in the male prison population, and to estimate the attrition rate of the study population over the follow-up period, in order to develop hypotheses and inform sample size calculations for future longitudinal research.

## Methods

### Sample

We carried out a 3-month prospective cohort study of new entrants to a large Category B men's prison in London, England. Category B prisons in general have a greater emphasis on security and less emphasis on training and rehabilitation than those in Category C and D, and a lesser emphasis on security than those in Category A. The study prison is a "local" prison housing prisoners taken directly from courts in the local area. The population of the prison is largely on remand (awaiting trial) but has sizeable minorities of sentenced prisoners and people detained under immigration law. The prison consists of a majority of double-occupancy cells and a minority of single-occupancy cells. Previous longitudinal research at the same study site documented a higher-than-average prevalence of self-harm behaviour making it well suited in terms of feasibility for a pilot study [14]. We chose a short follow-up period of three months due to the high attrition rates inevitably found in a remand prison.

The study population consisted of all prisoners entering the study prison between 17 March and 16 July 2022. Potentially eligible subjects were identified from prison reception lists via the National Offender Management Information System (NOMIS). A series of simple random samples of new arrivals was taken during each week of recruitment, totalling 675 across the duration of the study (see Fig 1). 147 of those sampled did not meet the inclusion criterion that their current prison spell had lasted less than 30 days at the time of arrival into the study prison. Screening procedures included a review of NOMIS records and, where needed, an additional in-person meeting. 218 potential participants were excluded due to no longer being in the prison (17/218), having an expected release date before the end of follow-up (143/218), having insufficient English language to participate (51/218), being too unwell to participate or having been transferred to hospital (4/218) or having a duplicate, incomplete or unavailable prison record (3/218). This left 310 eligible subjects, who were approached on prison wings to obtain informed consent. 152 participants were recruited to the study, whilst a further 32 agreed in principle to participate but were transferred or released before giving written consent. Following a baseline assessment interview, 3 participants withdrew their consent leaving a baseline cohort of 149 participants. Participants were followed up for a mean duration of 73.5 days (SD 24.1), with 83 participants (55.7%) completing the 3-month follow-up period. 66 participants (44.3%) were lost-to-follow-up due to transfer to another prison (35/66) or release (31/66). Four participants completed records-based follow-up but did not participate in the adjunct exit interview. As this was an exploratory pilot study, the sample size was determined by available resources rather than by a power calculation.

Ethical approval included permission to access non-identifiable demographic data (ethnicity and age) of non-participants, allowing for comparison of those who participated (n = 149) and those who were eligible but declined participation or withdrew consent (n = 129). There were no identifiable differences between these groups in terms of ethnicity ($X^2$ = 0.111, df = 1, p = 0.74) or calendar age ($t$ = 0.094, p = 0.93).

### Measures and procedures

The current study was part of the wider Self-Harm And Prison Environment Pilot (SHAPE-P) mixed methods project which included a qualitative study using interviews of self-harming prisoners and focus groups of prison staff to explore the relationship between aspects of the prison environment and self-harm. Putative environmental risk factors for the current study were identified by scoping the findings of two recent literature reviews [4,8], and from preliminary findings of the parallel qualitative study which suggested that

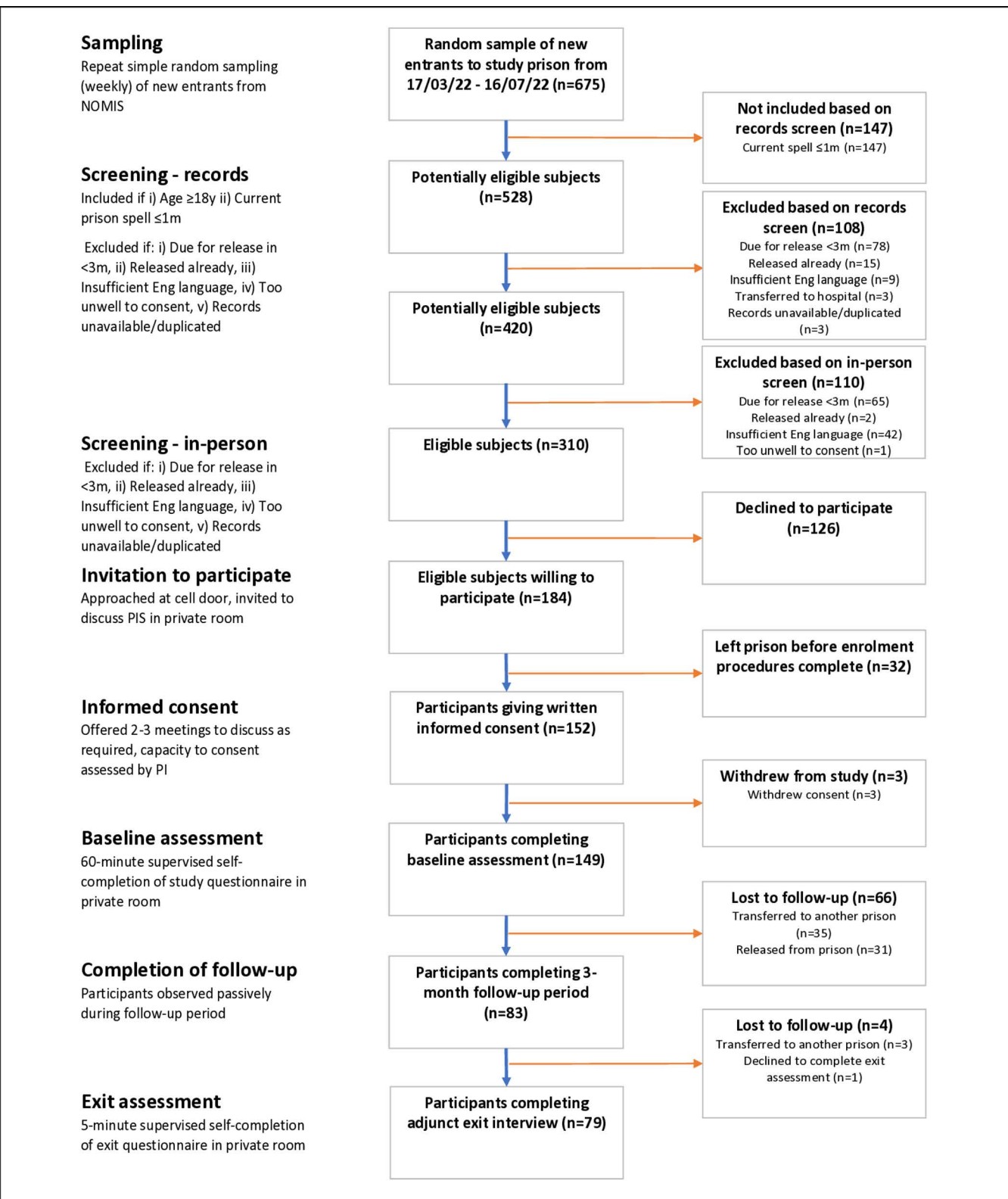

**Fig 1. Study flow chart.**

low staffing numbers, staff non-responsivity to distress, restrictive regime characteristics, vape replacement issues and COVID-19 prevention measures contribute to self-harm (in preparation [15]). Operational definitions for a total of 15 exposure factors were refined iteratively by the study team alongside scoping of exposure data availability and format at the study site. These included measures of prisoner activity (time in activities and work status), disciplinary measures (incentive level), placement-related factors (solitary confinement placement, single cell placement, cumulative number of cellmates, cumulative number of cell changes), contact with the outside world (social visit status, days to first phone call, time on phone), measures of staff and peer support worker numbers (staff-prisoner ratio, peer listener-prisoner ratio), staff responsivity (emergency bell response rate) and miscellaneous measures (vape use and undergoing COVID-19 in-cell isolation). Vape use was included and considered as an environment-related variable given that participants in the authors' parallel qualitative study cited frequent vape replacement issues as a contributing factor towards self-harm behaviour.

Three exposure variables (emergency bell response rate, staff-prisoner ratio and peer listener-prisoner ratio) were assessed by linking local group-level data for each prison wing with participants' starting location for each study week. Running weekly mean values were calculated from daily data for each variable. Measures of phone use were assessed using a British Telecom prison phone database. We identified COVID-19 isolation during follow-up from a local study prison dataset. Vape use was assessed during the baseline interview. All other exposures were measured using routinely collected data on participants' NOMIS prison record during follow-up. The primary study outcome was any self-harm behaviour during the 3-month follow-up, assessed from both participants' prison medical record and by semi-structured interview at end of follow-up. The outcome was double-assessed to test the reliability of records-based assessment for future research. Self-harm was defined as any intentional self-poisoning or injury irrespective of the apparent purpose of the act [16]. The secondary outcome was any self-harm ideation during follow-up, assessed by semi-structured interview at the end of follow-up. Due to limited resources, the timing of exposures and outcomes was not assessed in this study. Table 1 details operational definitions and data characteristics for all study variables.

Participants took part in a total of 1–3 meetings with researchers, as required, before giving written consent. Verbal consent was documented for those without sufficient literacy skills (1/149). All screening procedures were carried out by TS, an experienced forensic psychiatrist, who assessed capacity to consent. Subsequent consent and baseline interview procedures were carried out either by TS or by IH, a Masters level clinical psychologist.

Recruitment and interview procedures were carried out within six weeks of arrival into custody (range 4–42 days, *M* 25.6 days). A baseline interview included assessment of legal status on entry, historical violent offending, current psychiatric diagnosis and historical self-harm outside prison as part of a 30-item SHAPE study questionnaire (see S1 Questionnaire). The interview lasted approximately one hour and included other questionnaire instruments assessing a range of self-harm risk factors for the purpose of a separate study within the wider SHAPE project (in preparation [17]). Other demographic data, including age and ethnicity category, were collected from the NOMIS record.

The exit questionnaire was carried out within approximately 4 weeks of completion of 3-month follow-up (range 0–31 days, *M* 7.7 days). Wording for the 5 questionnaire items adapted items from the Self-Injurious Thoughts and Behaviours Interview (SITBI) [18], to account for the broader definition of self-harm used in this study as described above (see S2 Questionnaire). Researchers were blinded to all participants' exposure values except for vape use (which was assessed during baseline interviews) at the time of carrying out the exit

**Table 1. Operational definitions and data characteristics for exposure, confounder and outcome variables.**

| Exposure variable | Definition | Units | Data type | Data source | Completeness in % (count/total) |
|---|---|---|---|---|---|
| Time in activities | Total time in any prison activity, including courses and education but excluding work | Days | Continuous | NOMIS | 100 (82/82) |
| Working status | Any days in an allocated job in prison | – | Binary | NOMIS | 100 (82/82) |
| Incentive level[1]* | Prisoner incentive level during follow-up (Standard-throughout, Ever-enhanced, Ever-basic) | Category | Categorical | NOMIS | 92.7 (76/82) |
| Solitary confinement | Any placement in prison's Care & Separation Unit (solitary confinement) during follow-up | – | Binary | NOMIS | 100 (82/82) |
| Single cell[2] | Any placement in a single occupancy cell | – | Binary | NOMIS | 95.1 (78/82) |
| Cell changes* | Total number of cell changes | – | Continuous | NOMIS | 100 (82/82) |
| Cellmates* | Total number of cellmates | – | Continuous | NOMIS | 94.9 (74/78) |
| Social visit | Receipt of any social visit | – | Binary | NOMIS | 86.6 (71/82) |
| Time to first phone call[3]* | Time from prison entry to first phone call using permanent PIN | Days | Continuous | Prison phone database | 96.3 (79/82) |
| Time on phone* | Total time spent on phone using permanent PIN (excluding phone calls lasting less than 30 seconds) | Hours | Continuous | Prison phone database | 57.3 (47/82) |
| Vape use* | Any current vape use at time of baseline assessment (self-report) | – | Binary | Self-report (baseline) | 98.7 (81/82) |
| COVID-19 isolation[6]* | Any individual COVID-19 isolation (whether originating from participant or cellmate infection) | – | Binary | Local prison data | 100 (82/82) |
| Emergency bell response rate[4]* | Daily proportion of emergency bell calls that are answered within target time of 5 minutes (assessed at wing-level, linked to participants' location) | % | Continuous | Local prison data | 100 (214/214) |
| Staffing ratio | Daily mean staffing number across morning and evening shifts, per wing capacity (assessed at wing-level, linked to participants' location) | Ratio | Continuous | Local prison data | 99.5 (213/214) |
| Prison Listener ratio[5] | Daily mean Prison Listener number, per wing capacity (assessed at wing-level, linked to participants' location) | Ratio | Continuous | Local Samaritans data | 100 (214/214) |
| **Confounder variable** | | | | | |
| Age | | Years | Continuous | NOMIS | 100 (82/82) |
| Ethnicity | National Offender Management Information System (NOMIS) ethnicity category: White, Black, Asian, Mixed, Other | Category | Categorical | NOMIS | 100 (82/82) |
| Violence history | Defined as any previous conviction for a violent offence | – | Binary | Self-report | 100 (82/82) |
| **Outcome variable** | | | | | |
| Self-harm episode | Any episode of self-harm behaviour | – | Binary | Prison medical record (Systm1) | 100 (82/82) |
| | | | | Self-report (exit interview) | 95.1 (78/82) |
| Self-harm ideation | Any thoughts of self-harm behaviour | – | Binary | Self-report (exit interview) | 95.1 (78/82) |

[1]The Incentive and Earned Privilege Scheme (IEPS) is a behavioural incentive programme used across prisons in England and Wales. New prisoners start at 'standard' level. Downgrading to 'basic' for results in loss of earnable incentives including extra and improved visits; eligibility to earn higher rates of pay; access to in-cell television; opportunity to wear own clothes; access to private cash and additional time out of cell. It is separate from the disciplinary system.

[2]Double cell occupancy is standard in the study prison. Single cell occupancy in 'ordinary location' (i.e., outside of solitary confinement or healthcare wing) can result from different scenarios including high violence risk rating by the prison, medical reasons (for example, severe incontinence or severe PTSD) or temporarily whilst awaiting a new cellmate.

[3]New prisoners are allocated a Personal Identification Number which is used to make out-going phone calls from the cell. A contact must be approved by the prison prior to the first phone call.

[4]Emergency cell bells are available in each prison cell for prisoners to seek urgent attention from an officer. The target response time in the study prison is 5 minutes.

[5]The Listener scheme is a peer-support scheme within prisons run by Samaritans, a large suicide prevention charity. Listeners are trained by volunteers to provide confidential emotional support for peers who are struggling to cope or feeling suicidal.

[6]In the study prison, isolation protocols in place during the study period stated that in cases of any confirmed COVID-19 infection amongst occupants of a double occupancy cell, both occupants should isolate for 10 days.

* These variables were included based on preliminary findings from parallel qualitative work.

interviews. The end of follow-up was defined as the end of participants' third month in prison and no data was extracted from their records after this point. At the exit interview, participants were asked about outcomes and exposures occurring during the follow-up period only (i.e., any self-reported outcome occurring after the end of three months was discounted). Due to expected low literacy levels in the study population, all questionnaires were administered in interview format by TS and IH.

Ethical and HMPPS regulatory approval for the study were given by an NHS Research Ethics Committee (22/WA/0007) and HM Prison and Probation Service (2022-009). All researchers were independent of the regular prison staff.

## Statistical analysis

Those lost-to-follow-up (66/149) were excluded from analyses. We additionally excluded one further participant with a period of placement in the healthcare wing from all analyses because of its markedly different environment which resulted in outlier values for many exposure variables including, for example, staffing ratio. The final prospective cohort thus consisted of 82 participants. One variable, total time on phone, was excluded from the analysis due to a high proportion of missing values (42.7%).

We categorised one exposure variable – incentive level – where all participants fell into one of three categories (ever basic, standard throughout, ever enhanced). Participants with missing data for individual-level exposures were excluded from each analysis (see data completeness, Table 1). One missing group-level exposure data point for staff-prisoner ratio was imputed by calculating the mean of values for the previous and forthcoming weekdays. Reliable occupancy data were not available to calculate staff-prisoner ratio and peer listener-prisoner ratio. Therefore, total capacity was used as a proxy assuming 100% occupancy throughout the study. One prison wing was closed at the start of the study but gradually reopened during the second half of the study and reached approximately 50% capacity at end of the follow-up period. The total occupancy for this particular wing was imputed as 25% of capacity throughout the period of reopening based on consultation with prison officials.

We described the data using frequencies and proportions for categorical variables, means and standard deviations for normally distributed numerical variables, and medians and inter-quartile ranges for discrete and skewed numerical variables.

Odds ratios were calculated for the association of each prison-specific exposure with both study outcomes using separate univariable (unadjusted) and multivariable (adjusted) logistic regression models in R. Due to low- or zero-counts for both study outcomes, we did not carry out analyses for solitary confinement as a predictor.

Adjustments were made for participant calendar age, ethnicity (white/non-white), and violent offending history. These factors were pre-selected as potential confounders based on their theoretical impact on both exposure and outcome. In a logistic regression model containing only potential confounders and the study outcomes, white ethnicity was found to strongly predict self-harm episode (OR 11.41, CI 2.07–213.73, p = 0.02) but not self-harm ideation (OR 1.97, CI 0.66–6.42, p = 0.23), whilst no significant associations were seen between age or violent offending history and either outcome. A p-value cut-off of < 0.05 was used to designate statistical significance for all tests.

We carried out sensitivity analyses for outlier values, defined as > 3 standard deviations from the mean. Outliers existed for three continuous variables, number of cell changes (2/82), number of cellmates (1/82) and staff-prisoner ratio (1/82). A separate sensitivity analysis was carried out for all predictors by removing participants with any placement in solitary confinement (2/82) from all analyses.

## Results

### Cohort characteristics

Cohort demographics are described in Table 2. Participants (N = 149) were all men with a median age of 34 (*IQR* 27–43). The sample was representative in age ($X^2$ = 2.82, df = 3, p = 0.42) and ethnicity ($X^2$ = 3.74, df = 4, p = 0.44) of the wider population at the study prison based on data from 2021 [19–20]. Those who completed follow-up were comparable to those lost-to-follow-up except that they were more likely to have been on remand at time of baseline assessment. The majority of participants were on remand at time of entry to the study (111/149; 74.5%) with minorities who were sentenced prisoners (25/149; 16.8%), recalled on license (6/149; 4.0%) or of other legal status including detained under immigration law (7/149; 4.7%). Amongst those completing follow-up, the proportion on remand was higher (72/82; 87.8%), in large part because a larger proportion of prisoners who were already sentenced on entry to the

Table 2. Cohort characteristics at beginning and end of study compared with those declining to participate.

| Characteristic | Final cohort, n = 82 (%) | Cohort sub-groups according to outcome | | | |
|---|---|---|---|---|---|
| | | Self-harm behaviour n = 14 (%) | No self-harm behaviour n = 68 (%) | Self-harm ideation n = 22 (%) | No self-harm ideation n = 56 (%) |
| **Age (y)** | | | | | |
| *Mdn* | 34.0 | 33.5 | 34.0 | 32.0 | 34.0 |
| *IQR* | 27–43.75 | 26.5–42.5 | 27–44 | 26–37 | 27–44 |
| *M* | 36.28 | 35.57 | 36.43 | 33.41 | 36.68 |
| 18–29 | 30 (36.6) | ≥5 | 24 (35.3) | ≥5 | 20 (35.7) |
| 30–44 | 35 (42.7) | ≥5 | 30 (44.1) | ≥5 | 24 (42.9) |
| 45+ | 17 (20.7) | ≤5 | 14 (20.6) | ≤5 | 12 (21.4) |
| **Ethnicity** | | | | | |
| White | 48 (58.5) | ≥9 | 35 (51.5) | 16 (72.7) | 31 (55.4) |
| Non-white | 34 (41.5) | ≤5 | 33 (48.5) | 6 (27.3) | 25 (44.6) |
| Asian | ≥5 | ≤5 | ≥5 | ≤5 | ≥5 |
| Black | ≥5 | ≤5 | ≥5 | ≤5 | ≥5 |
| Mixed | ≤5 | ≤5 | ≤5 | ≤5 | ≤5 |
| Other | ≥5 | ≤5 | ≥5 | ≤5 | ≥5 |
| **Violent offending** | | | | | |
| Yes | 38 (46.3) | 9 (64.3) | 29 (42.6) | 14 (63.6) | 21 (37.5) |
| No | 44 (53.7) | 5 (35.7) | 39 (57.4) | 8 (36.4) | 35 (62.5) |
| **Legal status on entry** | | | | | |
| Remand | 72 (87.8) | ≥9 | 60 (88.2) | ≥17 | 47 (83.9) |
| Non-remand | 10 (12.2) | ≤5 | 8 (11.8) | ≤5 | 9 (16.1) |
| Sentenced | ≤5 | ≤5 | ≤5 | ≤5 | ≤5 |
| Recalled | ≤5 | ≤5 | ≤5 | ≤5 | ≤5 |
| Other | ≤5 | ≤5 | ≤5 | ≤5 | ≤5 |
| **Current mental disorder** | | | | | |
| Yes | 29 (35.4) | ≤5 | 18 (26.5) | 16 (72.7) | 11 (19.6) |
| No | 53 (64.6) | ≥9 | 50 (73.5) | 6 (27.3) | 45 (80.4) |
| **Previous self-harm outside prison** | | | | | |
| Yes | 20 (24.4) | ≥9 | 8 (11.8) | 12 (54.5) | 6 (10.7) |
| No | 62 (75.6) | ≤5 | 60 (88.2) | 10 (45.5) | 50 (89.3) |

study were lost to follow-up (21/25, 84.0%) than those who started on remand (39/111, 35.1%) due to transfer to other prisons. No difference was observed in the proportion of participants of other legal statuses in the final cohort compared to baseline. A previous conviction for violent offence (68/149, 46.3%) and an active diagnosis or treatment for any mental disorder at baseline interview (59/149, 39.9%) were self-reported by large minorities of the initial cohort, respectively. 26.9% (40/149) of participants reported a history of self-harm outside of prison.

The event rate for any self-harm behaviour during follow-up was 17.1% (14/82) based on medical records and 16.7% based on self-report (13/78), with discordance between the two measures in 5 of 78 (6.1%) cases for whom data was available such that 19.5% of participants (16/82) had a positive classification for self-harm behaviour on at least one measure. The event rate for any self-reported self-harm ideation was 28.2% (22/78). The concordance between self-harm behaviour outcome from records and self-harm ideation outcome from self-report was 82.1% (64/78) amongst those for whom data was available for both measures. No participants died by suicide during follow-up.

Outcome and predictor values for all predictors are presented in Table 3. Outcome counts are presented for categorical and binary exposure variables, whilst median and interquartile range values are presented for skewed numerical exposures and mean and standard deviation values are presented for normally distributed exposures. During the follow-up period, 29.3% (24/82) of the final cohort had any work, whilst 17.1% (14/82) received a down-grading to basic incentive level. 19.5% (16/82) experienced placement in a single occupancy cell. 30.5% (25/82) received one or more social visits. Participants waited a median duration of 6 days to make their first phone call out of the prison. The cohort took part in low levels of purposeful activity during follow-up (median 2 days) and spent time in locations with a mean ratio of one staff member per 31.88 prisoners.

## Regression analyses

Unadjusted and adjusted odds ratios for logistic regression models are shown in Table 4. More frequent cell changes (OR 1.83, CI 1.28–2.86, p = 0.003), a higher number of different cell-mates (OR 1.52, CI 1.14–2.17, p = 0.009), and any placement in a single cell (OR 4.31, CI 1.06–18.24, p = 0.04) were significantly associated with self-harm behaviour in adjusted models. Placement in areas with a higher number of prisoners per member of staff was significantly associated with reduced self-harm behaviour in adjusted models (OR 0.89, CI 0.78–0.99, p = 0.04). There was no significant relationship between self-harm behaviour and other variables such as time in activities, non-working status, a lack of social visits, time to first phone call, vape use or COVID-19 isolation.

Single cell placement (OR 3.54, CI 1.05–12.45, p = 0.04) was associated with self-harm ideation in adjusted logistic regression models. No relationship was identified with the remaining variables and self-harm ideation.

## Sensitivity analyses

The significant association between more frequent cell changes and self-harm behaviour remained after excluding two high outlier values. However, the association between more frequent cellmate changes and self-harm behaviour was no longer significant after excluding one high outlier value (OR 1.33, CI 0.96–1.96, p = 0.11, n = 81). The association seen for higher prisoner-staff ratio and self-harm behaviour was no longer significant after excluding one low outlier value (OR 0.97, CI 0.83–1.15, p = 0.73, n = 81).

After excluding participants with any placement in solitary confinement, the association seen between self-harm behaviour and single cell placement (OR 3.01, CI 0.63–13.71, p = 0.15,

**Table 3. Study exposures and self-harm outcome frequencies.** Data on distribution of exposure values between self-harm ideation outcome groups for participants not presented where outcome value is missing (4/82).

| Exposure (units) | Final cohort, n = 82 (%) | Self-harm behaviour n = 14 (%) | No self-harm behaviour n = 68 (%) | Self-harm ideation n = 22 (%) | No self-harm ideation n = 56 (%) |
|---|---|---|---|---|---|
| **Working status** | | | | | |
| Not working | 58 (70.7) | ≥9 | 47 (69.1) | 14 (63.6) | 40 (71.4) |
| Any work | 24 (29.3) | ≤5 | 21 (30.9) | 8 (36.4) | 16 (28.6) |
| **Incentive level** | | | | | |
| Ever enhanced | 23 (28.1) | ≤5 | 21 (30.1) | 5 (22.7) | 17 (30.4) |
| Standard | 39 (47.6) | ≥5 | 33 (48.5) | 9 (40.9) | 28 (50.0) |
| Ever basic | 14 (17.1) | ≥5 | 9 (13.2) | 7 (31.8) | 7 (12.5) |
| Missing data | 6 (7.3) | 1 | 5 (7.4) | 1 (4.6) | 4 (7.1) |
| **Solitary confinement** | | | | | |
| Yes | ≤5 | ≤5 | ≤5 | ≤5 | ≤5 |
| No | ≥77 | ≥9 | ≥63 | ≥17 | ≥51 |
| **Single cell placement** | | | | | |
| Yes | 16 (19.5) | 6 (42.9) | 10 (14.7) | 8 (36.4) | 8 (14.3) |
| No | 62 (75.6) | 7 (50.0) | 55 (80.9) | 13 (59.1) | 46 (82.1) |
| Missing data | 4 (4.9) | 1 (7.1) | 3 (4.4) | 1 (4.6) | 2 (3.6) |
| **Social visit status** | | | | | |
| No social visit | 25 (30.5) | 3 (21.4) | 22 (32.4) | 7 (31.8) | 18 (32.1) |
| Any social visit | 46 (56.1) | 5 (35.7) | 41 (60.3) | 10 (45.5) | 34 (60.7) |
| Missing data | 11 (13.4) | 6 (42.9) | 5 (7.4) | 5 (22.7) | 4 (7.1) |
| **Current vape use (at baseline)** | | | | | |
| Yes | 64 (78.1) | ≥9 | 53 (77.9) | ≥16 | 42 (75.0) |
| No | 17 (20.7) | ≤5 | 14 (20.6) | ≤5 | 14 (25.0) |
| Missing data | 1 (1.2) | 0 | 1 (1.5) | 1 | 0 (0.0) |
| **COVID-19 isolation** | | | | | |
| Yes | 8 (9.8) | ≤5 | 7 (10.3) | ≤5 | 5 (8.9) |
| No | 74 (90.2) | ≥9 | 61 (89.7) | ≥17 | 51 (91.1) |
| **Time in activities (days)** | | | | | |
| Mdn | 2 | 5 | 2 | 2 | 2 |
| IQR | 1–8 | 2–8.5 | 1–8 | 1–7 | 1–8.5 |
| **Cell changes (cumulative)** | | | | | |
| Mdn | 3 | 5 | 3 | 3 | 3 |
| IQR | 3–5 | 2.25–8 | 3–4 | 2–4.75 | 3–5 |
| **Cellmates (cumulative)** | | | | | |
| Mdn | 3 | 4 | 3 | 4 | 3 |
| IQR | 3–4 | 3.25–5.75 | 2.75–4 | 2.25–5 | 3–4 |
| Missing data | 4 (4.9) | 0 (0.0) | 4 (5.9) | 0 (0.0) | 0 (0.0) |
| **Time to first phone call (days)** | | | | | |
| Mdn | 6 | 5 | 7 | 6 | 6 |
| IQR | 2–11 | 1–15.5 | 2–11 | 2–16 | 1–11 |
| **Total time on phone (hours)** | | | | | |
| Mdn | 27.0 | 49.9 | 26.1 | 28.7 | 27.6 |
| IQR | 10.4–67.5 | 25.1–70.5 | 10.4–67.5 | 13.1–42.9 | 18.4–91.0 |
| Missing data | 35 (42.7) | 10 (71.4) | 25 (36.8) | 12 (54.5) | 20 (35.7) |

*(Continued)*

**Table 3.** (Continued)

| Exposure (units) | Final cohort, n = 82 (%) | Self-harm behaviour n = 14 (%) | No self-harm behaviour n = 68 (%) | Self-harm ideation n = 22 (%) | No self-harm ideation n = 56 (%) |
|---|---|---|---|---|---|
| **Prisoner-Listener ratio** | | | | | |
| *Mdn* | 82.2 | 71.9 | 82.7 | 77.3 | 82.5 |
| *IQR* | 61.4–103.0 | 58.0–92.6 | 65.6–108.9 | 61.6–92.7 | 52.1–104.2 |
| **Prisoner-staff ratio** | | | | | |
| *M* | 31.88 | 29.17 | 32.44 | 31.56 | 31.86 |
| *SD* | 5.31 | 8.26 | 4.37 | 7.07 | 4.69 |
| **Emergency bell response rate (%)** | | | | | |
| *M* | 69.10 | 72.37 | 68.42 | 70.68 | 68.74 |
| *SD* | 7.84 | 6.63 | 7.95 | 6.48 | 8.32 |

n = 80) was no longer significant. No other associations with self-harm behaviour were significantly affected by excluding such participants. The association between single cell placement and self-harm ideation remained significant after excluding participants with any placement in solitary confinement. A sensitivity analysis excluding five participants for whom wing-based staffing data was imputed (5/82) did not change the observed association between a higher prisoner-staff ratio and reduced self-harm behaviour (OR 0.87, CI 0.75–0.97, p = 0.02, n = 77).

## Discussion

In this study of 83 prisoners, we tested 15 measures of environmental risk factors with self-harm over a 3-month period in one male remand prison in England with a higher-than-average prevalence of self-harm. We found that more frequent changes of cell and cellmate and single cell placement were associated with a higher risk of self-harm behaviour. Placement in areas with a higher number of prisoners per staff member was associated with a lower risk of self-harm behaviour.

A recent meta-analysis of case-control and cross-sectional evidence suggested that custody-specific factors may confer increased risk of self-harm [4]. These included aspects of prisoners' relationships with staff and peers (disciplinary infractions, sexual or physical victimisation, being threatened with violence), cell placement (solitary confinement), and a lack of work, activities and social contact or visits in prison. However, we are not aware of longitudinal studies that have examined such relationships, except in the cases of solitary confinement [21–23] and disciplinary infractions [23,24]. This exploratory pilot prospective cohort study is the first empirical study, to our knowledge, that examines the relationship between self-harm and incentive level, number of cellmates, measures of in-cell phone use, and measures of prison staffing level, staff responsiveness to prisoners and availability of peer listeners. It is also the first longitudinal study to examine the relationships between self-harm behaviour and including single cell placement, cell changes, and measures of work and activities.

The association seen between self-harm and frequent cell changes accords with findings from one case-control study in which high housing (prison cell) moves in the previous two years was associated with self-harm [6]. Our finding regarding single cell placement and self-harm behaviour is in keeping with the findings of a recent meta-analysis [4] which identified a similar sized but non-significant association (OR 1.5, 95% CI 0.8–2.9, n = 4309, p = 0.23) based on case-control and cross-sectional data. The association between self-harm behaviour and greater number of cellmates is novel, and similar in size to that seen for cell changes.

**Table 4. Odds ratios of predictors for self-harm episode and ideation in unadjusted and adjusted logistic regression models.**

| Theme | Exposure | Self-harm behaviour (n = 14/82) | | | | Self-harm ideation (n = 22/78) | | | |
| --- | --- | --- | --- | --- | --- | --- | --- | --- | --- |
| | | Unadjusted model | | Adjusted model[1] | | Unadjusted model | | Adjusted model[1] | |
| | | OR (95% C.I.) | p | OR (95% C.I.) | p | OR (95% C.I.) | p | OR (95% C.I.) | p |
| Regime | Time in activities (days) | **1.02** (0.97–1.07) | 0.32 | **1.01** (0.96–1.06) | 0.65 | **1.00** (0.95–1.05) | 0.94 | **1.00** (0.94–1.04) | 0.85 |
| | **Working status** | | | | | | | | |
| | No work | **1.64** (0.45–7.79) | 0.48 | **1.05** (0.24–5.55) | 0.95 | **0.70** (0.25–2.04) | 0.50 | **0.50** (0.15–1.63) | 0.25 |
| | Any work | 1.00 | – | 1.00 | – | 1.00 | – | 1.00 | – |
| | **Incentive level** | | | | | | | | |
| | Ever enhanced | **0.52** (0.07–2.52) | 0.45 | **0.64** (0.08–3.63) | 0.63 | **0.92** (0.25–3.12) | 0.89 | **0.88** (0.22–3.30) | 0.85 |
| | Standard | 1.00 | – | 1.00 | – | 1.00 | – | 1.00 | – |
| | Ever basic | **3.06** (0.74–12.63) | 0.12 | **5.03** (0.88–34.93) | 0.08 | **3.11** (0.86–11.65) | 0.08 | **2.76** (0.66–12.37) | 0.17 |
| Cell | **Single cell placement** | | | | | | | | |
| | Yes | **4.71** (1.29–17.36) | 0.02 | **4.31** (1.06–18.24) | 0.04 | **3.54** (1.11–11.52) | 0.03 | **3.54** 1.05–12.45) | 0.04 |
| | No | 1.00 | – | 1.00 | – | 1.00 | – | 1.00 | – |
| | **Number of cell changes (cumulative)** | **1.48** (1.15–2.01) | 0.006 | **1.83** (1.28–2.86) | 0.003 | **1.09** (0.87–1.35) | 0.42 | **1.10** 0.89–1.40) | 0.40 |
| | **Number of cellmates (cumulative)** | **1.31** (1.03–1.75) | 0.04 | **1.51** (1.14–2.17) | 0.009 | **1.25** (1.01–1.63) | 0.06 | **1.25** (1.00–1.64) | 0.07 |
| Contact with outside world | **Social visit status** | | | | | | | | |
| | No social visit | **1.12** (0.21–5.00) | 0.89 | **0.80** (0.14–4.04) | 0.78 | **1.32** (0.42–4.05) | 0.63 | **1.22** (0.35–4.14) | 0.75 |
| | Any social visit | 1.00 | – | 1.00 | – | 1.00 | – | 1.00 | – |
| | **Time to first phone call (days)** | **1.00** (0.96–1.04) | 0.80 | **1.00** (0.96–1.04) | 0.92 | **1.01** (0.97–1.03) | 0.69 | **1.01** (0.97–1.04) | 0.68 |
| Other | **Vape use (baseline)** | | | | | | | | |
| | Yes | **0.97** (0.26–4.70) | 0.96 | **0.66** (0.14–3.62) | 0.61 | **2.00** (0.57–9.44) | 0.32 | **1.33** (0.33–6.72) | 0.70 |
| | No | 1.00 | – | 1.00 | – | 1.00 | – | 1.00 | – |
| | **COVID-19 isolation** | | | | | | | | |
| | Yes | **0.67** (0.03–4.25) | 0.72 | **0.55** (0.03–4.09) | 0.61 | **1.02** (0.14–5.17) | 0.98 | **0.89** (0.11–5.07) | 0.90 |
| | No | 1.00 | – | 1.00 | – | 1.00 | – | 1.00 | – |
| Staff | **Prisoner-listener ratio** | **0.99** (0.98–1.01) | 0.25 | **0.98** (0.96–1.00) | 0.09 | **1.00** (0.99–1.01) | 0.73 | **1.00** (0.98–1.01) | 0.57 |
| | **Prisoner-staff ratio** | **0.90** (0.81–1.00) | 0.04 | **0.89** (0.78–0.99) | 0.04 | **0.99** (0.90–1.09) | 0.83 | **0.98** (0.89–1.08) | 0.71 |
| | **Emergency bell response rate** | **1.06** (0.99–1.15) | 0.10 | **1.09** (1.00–1.20) | 0.06 | **1.03** (0.97–1.10) | 0.33 | **1.03** (0.96–1.10) | 0.39 |

[1]Adjusted for age, ethnicity and violent offending history.

The finding that higher prisoner-staff ratio was associated with reduced odds of self-harm behaviour is novel.

The current study did not identify any significant associations between self-harm outcomes and time in activities, working status and social visit status. This is in contrast to previous studies which have identified associations between these variables and self-harm, including a large meta-analysis which addressed unemployment in prison (OR 1.9, 95% CI 1.5–2.5, n = 3311) and a lack of social contact or visits (OR 2.3, 95% CI 1.5–3.5, n = 2153)[4], and an ecological study of UK prisons which identified an association between prisons with lower time in purposeful activity and completed suicide amongst prisoners [25].

The attrition rate in this study (44.3%) was higher than that seen at 3-month follow-up in a prospective cohort study of sentenced prisoners at the same study prison (15.5%) [14]. This is likely due to higher turnover amongst participants who were on remand when recruited to the study.

## Limitations

The low total event counts in this study resulted in low precision of effect size calculations with wide confidence intervals in some places. Equally, results for cell changes, number of cellmates and staff-prisoner ratio were sensitive to the impact of outlier exposure values, and the result for single cell placement was sensitive to removal of the sub-group with solitary confinement placement.

Aggression is a plausible confounder of the relationship between prison environment factors and self-harm. Prisoner aggression typically determines single-occupancy cell placement as a result of prison cell-sharing risk assessments (which are concerned with the risk of inter-prisoner aggression, see Table 2 footnotes), as well as number of cell and cellmate changes. A recent meta-analysis identified violence/assault perpetration as being a potential risk factor for self-harm (OR 3.8, 95% CI 0.9–15.8, n = 276,968, p = 0.07) [4]. We controlled for previous violent offending but did not measure in-study aggression amongst participants. Unmeasured confounding from in-study aggression may explain the loss of significance of the relationship between single cell placement and self-harm behaviour when excluding the sub-group who also had solitary confinement placement. The direction of any bias introduced by unmeasured confounding is unclear, but its magnitude may have been reduced by adjusting for previous violent offending status given that this predicted self-harm ideation amongst the cohort.

The large proportion of participants lost-to-follow-up in this study (44.3%) raises the risk of selection bias in those completing follow-up. Reassuringly, completers were comparable in age and ethnicity category to the original cohort. Similarly, this study used records to measure exposures and outcomes, raising the risk of information bias. However, primary outcome assessment using prison medical records and self-report produced highly similar results. The study was conducted in a prison with a higher-than-average prevalence of self-harm, and this higher prevalence may be influenced by unmeasured systematic biases such as the characteristics of the prison environment. Such biases would lower the generalisability of findings to other settings and prison sub-populations.

The timing of exposure and outcome were not measured in this study, so it was not possible to demonstrate the direction of effect in the relationships identified between environmental factors and self-harm outcomes. Reverse causality may explain the association found between placement in areas with higher prisoner-staff ratios and reduced self-harm behaviour, given the common practice of monitoring prisoners who self-harm with regular or constant staff observation, resulting in a lower number of prisoners per member of staff for that area.

The small sample size prevented us from assessing the extent and impact of collinearity between predictors. High collinearity between multiple environment predictors is plausible, for example between participants' cell changes and total number of cellmates. We were also unable to assess whether any other known risk factors for self-harm, such as mental disorder, psychological distress and history of abuse mediated or moderated the relationships between study exposures and outcomes. One plausible explanation of the observed association between cell-related factors and self-harm behaviour is that this relationship is mediated by unmeasured individual differences in emotion dysregulation and/or aggression.

## Future research

The limitations above make the results of this study preliminary and in need of replication in better powered longitudinal studies. Basic incentive level and emergency bell response rate exposures warrant further examination in larger samples given that the results in this small sample approached the threshold for statistical significance. Larger samples will also permit

sub-group analyses. Future studies assessing time-to-event data could discern the direction of any effects underlying the associations observed. Replication of the study findings in research in prisons of varying security, location, function and with varying prevalence of self-harm, as well as in the female prison population, will improve generalisability and allow for cross-comparison of the strength of effects. Future research should seek to address in-study aggression and recent self-harm behaviour (for example, self-harm occurring in the period between initial arrest and imprisonment) as potential confounders of relationships between prison environment exposures and self-harming outcomes. Discordance between outcomes assessed via prison medical records and self-report was low (6.1%) but may nonetheless have implications for future study designs in this area.

We identified 12 other potentially relevant environment factors which could not be assessed in this study due to non-availability or poor quality of data including time out of cell, prison regime cancellations, bullying and victimisation and prison occupancy rate. Where feasible, future research should attempt to examine the relationships of such factors with self-harm.

## Clinical implications

If replicated, the relationships observed in this study between self-harm and several prison environment factors may have implications for practice in the assessment and management of self-harm by clinical and non-clinical staff. This may also have system-level implications for mental health promotion measures in prisons and for resources spent on intensive care planning processes such as the Assessment, Care in Custody and Teamwork (ACCT) process employed in prisons in England and Wales [26,27]. The association between single cell placement and self-harm also highlights the importance of other approaches prisons can take to reduce morbidity and mortality associated with self-harm, including – where appropriate – placement in safe (ligature-free) cells and the use of non-ligature clothing.

## Roles of researchers

TS was Chief Investigator for the project and was responsible with NB for study design. TS carried out all screening and initial approach procedures, counselled potentially eligible subjects about the study and assessed decision-making capacity. IH and CA were graduate level MSc psychology candidates; IH carried out baseline interviews and records data extraction alongside TS. TS, IH and CA carried out records data extraction. RS and SF provided advice on the statistical analysis, which TS carried out. NB provided overall supervision for the project. All authors contributed to the manuscript.

## Supporting information

**S1 Questionnaire. 30-item study questionnaire.**
(DOCX)

**S2 Questionnaire. 5-item exit questionnaire.**
(DOCX)

## Author contributions

**Conceptualization:** Thomas Stephenson, Nigel Blackwood.

**Data curation:** Thomas Stephenson, Imogen Harris, Charlotte Armstrong.

**Formal analysis:** Thomas Stephenson, Seena Fazel, Roxanna Short.

**Funding acquisition:** Thomas Stephenson, Nigel Blackwood.

**Investigation:** Thomas Stephenson, Imogen Harris, Charlotte Armstrong.

**Methodology:** Thomas Stephenson, Seena Fazel, Roxanna Short, Nigel Blackwood.

**Project administration:** Thomas Stephenson.

**Supervision:** Nigel Blackwood.

**Writing – original draft:** Thomas Stephenson.

**Writing – review & editing:** Imogen Harris, Charlotte Armstrong, Seena Fazel, Roxanna Short, Nigel Blackwood.

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
