## [Decision Letter · Decision Letter 0]

1 May 2024

PONE-D-24-04206Environmental risk factors for self-harm during imprisonment: a prospective cohort studyPLOS ONE

Dear Dr. Stephenson,

Thank you for submitting your manuscript to PLOS ONE. I have now received peer-review reports, and I am pleased to invite you to submit a revised version of the manuscript that addresses the points raised during the review process by Reviewer #1.

Their feedback was mainly provided to improve the rigour and transparency of reporting. Please do take their feedback into account, and if you find you are not able to do so, I would appreciate you detailing the reasons behind the decision, so that they can be considered following the submission of the revised manuscript.

We would be grateful if you could submit your revised manuscript by Jun 14 2024 11:59PM. If you need more time than this to complete your revisions, please reply to this message or contact the journal office at plosone@plos.org . Please include the following items when submitting your revised manuscript:

We look forward to receiving your revised manuscript.

Kind regards,

Dr Nasrul Ismail

Academic Editor

PLOS ONE

Journal Requirements:

Reviewers' comments:

Reviewer's Responses to Questions

**Comments to the Author**

1. Is the manuscript technically sound, and do the data support the conclusions?

Reviewer #1: Yes

Reviewer #2: Yes

2. Has the statistical analysis been performed appropriately and rigorously? 

Reviewer #1: Yes

Reviewer #2: Yes

3. Have the authors made all data underlying the findings in their manuscript fully available?

Reviewer #1: No

Reviewer #2: No

4. Is the manuscript presented in an intelligible fashion and written in standard English?

Reviewer #1: No

Reviewer #2: Yes

5. Review Comments to the Author

Reviewer #1: The authors have written an article regarding self-harm in prisoners. They have investigated environmental in-prison risk factors for self-harm in a prospective study. They initially included 149 prisoners within a remand setting who were assessed on a range of previously identified environmental risk factors for self-harm specific to the prison setting. The study is framed as a pilot study. During follow-up a substantial amount of participants were lost due to prison and sentencing related factors, which resulted in a final sample of 82 or 83 participants. Several of the investigated environmental risk factors were found to be associated with the outcome. The article contributes to the base of academic knowledge and highlights a public health problem within the prison system. The design of the study sound and acceptable in order to reach the authors’ aims. However, this article needs revision before it is ready for publication.

PLOSE ONE’s criteria for publication

1. This article contributes to the base of academic knowledge and presents results of original research by exploring prison environmental factors for self-harm and estimating their effects.

2. I have found no evidence of these findings being reported elsewhere, except in a preprint format, which adheres to PLOS ONE’s criteria for publication: https://www.medrxiv.org/content/10.1101/2024.01.31.24302059v1.

3. The statistical analyses were performed to a good technical standard but could require some improvements and the accompanying text requires improvements. This is elaborated on below.

4. Conclusions are generally presented in an appropriate fashion but due some concerns regarding the validity and robustness of the results, revisions are requested.

5. Generally, the article was written in good English but needs revision to improve clarity and reduce ambiguity.

6. The research appears to adhere to the ethical standards and it received approval from the relevant ethical board as well as the national prison and probation services.

7. The article generally follows appropriate reporting standards but requires some improvements. The data have not been made available to others, a decision that is sound given the design and participants of the study. Data requests are directed to relevant services.

Abstract and introduction

The authors summarize previous research and articulate their research questions. The introduction provides a concise understanding of the field, along with a rationale and aims for their study. It offers sufficient information about self-harm during imprisonment. However,

1. The introduction lacks contextualizing of self-harm during a life-course in antisocial, aggressive and/or offending populations. Self-harm is more likely to have occurred both before and after imprisonment. Furthermore, self-harm is overrepresented in acute phases (e.g., remand), acute psychiatric conditions (e.g., adjustment disorders), and generally enduring psychiatric disorders (e.g., anxiety, personality, and severe mental disorders). Thus, the introduction lacks information regarding self-harm as part of the psychiatric spectrum in prisoners and consequently lacks preceding information about psychiatric morbidity in prison populations. I believe this omission reflects a conscious choice from the authors. However, I would argue that this information is crucial in their introduction, and it seems the authors also consider it important, as evidenced, for example, by the inclusion of such details in their questionnaires.

2. The second sentence of the introduction, which mentions the troublesome doubling of self-harm incidents among prisoners. This substantial increase prompts questions about the report’s methodology and the reasons behind this trend. Furthermore, the citation of a non-academic article as the source raises concerns about the claim’s credibility (I have not reviewed the given reference). It would be beneficial to substantiate this claim with research from academic sources and from various contexts.

3. The final sentence of the first paragraph, which describes self-harm as a resource burden for service providers, lacks citations. Moreover, its relevance to the paper may be questioned. If the authors deem this information important to the introduction, further elaboration and supporting references are needed, but may also be saved for the discussion and clinical implications of the study.

4. The last sentence of the second paragraph requires improvements in language and clarity, along with more precise reporting of the odds ratio, including a confidence interval.

5. In the third paragraph, the authors discuss a wide array of prison environmental factors that might contribute to self-harm, but the direction of these associations should be clarified. Furthermore, while the authors note the current uncertainty around the strength of these effects, it is important to recognize that the strength of associations varies significantly depending on the context and prison population. This aspect warrants some elaboration.

6. The introduction incorporates multiple references to specifically female samples, which is not inherently problematic. However, given that this study focuses on an all-male sample, it is important to clearly highlight when employing such references.

7. The aims appear to be structured as a primary and a secondary aim, suggesting a hierarchy of importance. If my understanding is correct, this distinction should be explicitly stated.

8. The introduction provides minimal context regarding the secondary aim. This aim might be considered for removal. Alternatively, it would be beneficial to elaborate on the design of a wider array of previous studies (including sample size, and statistical approaches), and the insights they have offered for the purpose of this aim.

9. Mentioning that the article was part of a wider project is redundant for the aims paragraph. However, as a reader, I would appreciate a summary description of the SHAPE project (if this is what the authors are referring to here) in the methods section, but I leave this entirely up to the authors to decide.

Methods

1. Please describe the attrition, for example, how many participants were excluded due to their English language skills or decision-making capacity, respectively.

2. Participants lost due to suicide during the follow-up period should be mentioned, even if there were no cases, if this was measured.

3. Category B prisons are contrasted with Categories C and D but should also be contrasted with Category A.

4. A strength of this study is that the sample corresponds to the general prison population on key factors. However, note observed differences between prisoners awaiting trial, those sentenced, and those detained under immigration law in the final sample.

5. It is unclear whether the sample comprises "all male prisoners who had recently arrived" or an all-male sample.

6. Briefly describe the simple random sampling approach used in this study.

7. If I understand Figure 1, the sentence describing eligibility criteria in the sample segment is need of correction. Should it state “aged 18 or older, male, and current prison spell had lasted less than one month”?

8. The average length of follow-up was 73.5 days. Please indicate the standard deviation or other measure of variability.

9. The instruments used such as DUDIT and CTQ, are insufficiently described and not referenced. Furthermore, it is not clear how information from these instruments were utilized in this study and if they were administered alongside the supplemented questionnaires.

10. In the methods section, it is mentioned that this was an exploratory pilot study, a description with which I agree. However, the language in the abstract, aims, and discussion needs revision to consistently reflect this.

11. The three participants who withdrew their consent were not accounted for in the comparison of non-identifiable demographic data, as they seem to be missing from the stated n=126, referencing the last paragraph in the sample segment. This figure should likely be adjusted to 129 and comparative test rerun accordingly.

12. Throughout the document, ensure that statistical notation is used correctly, such as changing ‘mean’ to italicized M, and consistently reporting odds ratios with or without the 95% numeral in accordance to applied reporting standards.

13. The independence of the researchers from the regular prison staff should be clarified in the methods section. If there was no independence, this raises questions for the limitations section, such as the potential impact on informed consent and the data collected. Additionally, the lack of researcher blindness to participants could have influenced the findings. The potential effects of this potential lack of blinding and independence on the study’s reliability and impartiality need to be addressed.

14. SHAPE is used as an abbreviation but not explained.

15. The decision to include vape use as an exposure for self-harm warrants further explanation. Since this variable has been singled out based on the authors’ unpublished results, it is crucial to offer a rationale for its inclusion. Additionally, clarifying why vape use is considered an environmental risk factor, rather than a risk factor related to individual behaviour, is needed.

16. It is unclear whether number of cellmates includes a cumulative number of cellmates the participants have during the follow-up or the number of people the prisoner is sharing their cell with at data collection.

17. I would like confirmation that risk factors were assessed at baseline and not during the three-month follow-up period, as assumed from the prospective design, but this is not entirely clear.

18. The structure of variables such as the number of cell changes, number of cellmates, time to first phone call, time on the phone, emergency bell response rate, prisoner-staff ratio, and prisoner-listener ratio could be reconsidered, for example through standardization or dichotomization.

19. The methods state that TS and IH conducted interviews while the roles of researchers state TS, IH, and CA, correct or clarify this please.

20. Studies in preparation are generally referred to and referenced in the reference list.

21. Describe the methods of the scoping review and briefly describe the preliminary results from the parallel qualitative study you apply in this study.

22. Ethnicity is used both as a categorical variable in describing attrition and as a dichotomized variable in regression analysis. Please choose one approach for consistency.

23. The exit interview timing is unclear, as it is stated to be conducted after the end of the follow-up period, which is hard to conceptualize as to me the exit interview would represent the end of follow-up. This process was said to occur approximately 4 weeks after completion, in parenthesis it mentions 7 days. Please clarify the exact timing of the exit interviews and what actually represents the end of follow-up for consistency and understanding.

24. The authors state that they have included their secondary outcome, NSSI ideation, with a view to improving the statistical power. I disagree with this. Instead, they have studied two quite different outcomes (i.e. NSSI and NSSI ideation). In regards to power, the power would be increased by, for example, adding more participants to the study, the statistical tests employed, or changing the significance level – not by analysing a distinct outcome.

25. A (recent) history of self-harm and/or suicide attempts is captured in the baseline questionnaire and may represent a confounder for several analyses in this study, as mentioned in the limitations in regards to single cell and prisoner-staff ratio. This information could be used in sensitivity analyses.

26. In the light of the aims it is not clear what the 66 participants that dropped out during follow-up adds to the study. This should be clarified or reframe the study as a study of a sample of 83/82 individuals.

27. It is stated that the authors did not analyse solitary confinement as a predictor. However, such results are presented in Table 5.

28. The high levels of missingness in time on phone-variable require methods to deal with this issue or exclude it from analysis.

29. The authors state that they conduct multivariable logistic regression models. However, it seems to me as if the unadjusted models are univariable. This requires clarification.

30. It is confusing that the authors state that they use a set p-value cut-off, yet they consider a positive tendency (CI 0.93 – 7.75, p=0.071) to be associated with the outcome.

Figures and tables

1. Table 2 generally displays that those lost to follow-up differ in few aspects, aspects already mentioned in the results section. Mentioning this in text would suffice. Thus, removing some information from the table. This will improve its readability without losing any relevant information.

2. Tables 3 and 4 should be combined into a single table, with clear indications of the units for each variable (e.g., minutes, days, and percent). The table should, as currently presented, denote missing data. However, the accompanying text can then draw attention to the key findings and significant aspects of the data presented in this consolidated table rather than levels of missingness.

3. For readability, S3 Supplementary table 1 is better summarized in text and not as a supplement.

Results, discussion, and conclusions

1. Key information in regards to the validity of the secondary aim is the correlation or concordance or NSSI and NSSI ideation and could be reported. Auxiliary, the comparison between the 14/82 registered versus the 13/78 self-reported events of NSSI suggests a need to assess their concordance or discordance as suggested by the authors in the methods section. This also have bearing on the third limitation.

2. The baseline questionnaire lacks questions about the incidence of self-harm/suicide attempts during the ongoing legal process, focusing instead solely on occurrences within prison at any given time. Including this information is essential for a prospective study, an issue that is briefly mentioned in the limitations section.

3. I am concerned about the methodology used by the authors in treating the staff-to-prisoner ratio variable, specifically the imputation of numbers based on prison conditions in a new ward. This may have influenced the negative association found for prisoners per staff member. Without direct access to the data, it is impossible to evaluate this issue. Therefore, I recommend the authors to carefully consider this factor and potentially reanalyse their data. This concern is heightened by the observation that a previously significant relationship turned insignificant after the exclusion of just one outlier, casting further doubt on the validity of the results regarding the staff-to-prisoner ratio.

4. Remove the term "unexpected" in relation to the negative association found between staff-prisoner ratio and NSSI, as the article present no hypotheses and is framed as explorative.

5. Given the low number of participants and the relatively few positive primary outcomes, it is advisable to handle outliers through methods other than simple removal. Techniques such as Winsorization could be considered to explore or manage extreme values and its effects on the robustness of the results.

6. My interpretation is that the results does not indicate a major issue of power regarding interpretations of time in activities, working status and social visit status. An issue of power would have been indicative by a tendency in a given direction but with a too wide confidence interval to draw conclusive results. However, conceptually the issue of power is of course relevant here, as in any small sample study, but not based on the presented findings.

7. In the limitations section, you mention that the study size was too small to detect small to moderate effect sizes. Yet, you have identified some small to moderate effect sizes in the current study. This needs clarification.

8. Length of stay up to baseline could have had influence on the risk factors that was not accounted for and is a limitation. Furthermore, it is a limitation that information regarding the length of follow-up was not incorporated in statistical analyses and could be mentioned.

9. Given the authors discussion of aggression and single cell placements, your discussion, for example, in regards to the second limitation, would benefit from considering the context of existing studies on aggression (separate from self-directed aggression) as a factor linked to self-harm.

Thank you for an interesting read and good luck on your future endeavours. With revisions, I am confident that the current study, along with your upcoming larger study, will make significant contributions to the field. This is particularly relevant in a global context where prison populations are expanding, often without adequate consideration of the implications for individual and public health.

Reviewer #2: Thank you for the opportunity to review this paper which examines the relationship between environmental factors and self-harm in prison. More prospective studies of this kind are needed. The manuscript is well written and clear, the methods and variables well explained, and limitations acknowledged.

6. PLOS authors have the option to publish the peer review history of their article (what does this mean? ). If published, this will include your full peer review and any attached files.

**Do you want your identity to be public for this peer review?** For information about this choice, including consent withdrawal, please see our Privacy Policy .

Reviewer #1: **Yes: ** André Tärnhäll

Reviewer #2: No

---

## [Author Response · Author response to Decision Letter 1]

7 Jun 2024

Thomas.stephenson@kcl.ac.uk

1st June 2024

Dear Dr Ismail,

Thank you for the opportunity to revise our manuscript in relation to reviewer’s comments. Alongside our submission of a revised manuscript, in this letter we have outlined our response to each point raised by the reviewers. Our responses are in italics.

Points raised by reviewer #1

Abstract and introduction

1

The introduction lacks contextualizing of self-harm during a life-course in antisocial, aggressive and/or offending populations. Self-harm is more likely to have occurred both before and after imprisonment. Furthermore, self-harm is overrepresented in acute phases (e.g., remand), acute psychiatric conditions (e.g., adjustment disorders), and generally enduring psychiatric disorders (e.g., anxiety, personality, and severe mental disorders). Thus, the introduction lacks information regarding self-harm as part of the psychiatric spectrum in prisoners and consequently lacks preceding information about psychiatric morbidity in prison populations. I believe this omission reflects a conscious choice from the authors. However, I would argue that this information is crucial in their introduction, and it seems the authors also consider it important, as evidenced, for example, by the inclusion of such details in their questionnaires.

Response: We are grateful to the Reviewer for this observation and have amended the introduction to provide further acknowledgement of the importance of these factors which, we agree, are crucially important to understanding self-harm behaviour within these populations. Thus, we have included the following within the second paragraph of the Introduction: “These include self-harm behaviours prior to imprisonment and pre-existing psychiatric morbidity (for example, personality and depressive disorders). High levels of such morbidity have been documented across prison populations in many countries.” We have not, in this case, cited a link between remand status and self-harm behaviours as this association was not found in the recent meta-analysis we have cited.

2

The second sentence of the introduction, which mentions the troublesome doubling of self-harm incidents among prisoners. This substantial increase prompts questions about the report’s methodology and the reasons behind this trend. Furthermore, the citation of a non-academic article as the source raises concerns about the claim’s credibility (I have not reviewed the given reference). It would be beneficial to substantiate this claim with research from academic sources and from various contexts.

Response: We thank the Reviewer for this observation. In our view, the data referenced are credible because the UK Ministry of Justice have standardised data collection and reporting mechanisms across prisons in England and Wales that record self-harm incidence. There is little epidemiological data of comparable size available from non-governmental sources. We have additionally cited the last available findings on prevalence of self-harm amongst men and women in a national prevalence study across England and Wales based on data from 2004-2009 in the first sentence of the Introduction section, in support of the claim that self-harm is a major public health issue in prisons.

3

The final sentence of the first paragraph, which describes self-harm as a resource burden for service providers, lacks citations. Moreover, its relevance to the paper may be questioned. If the authors deem this information important to the introduction, further elaboration and supporting references are needed, but may also be saved for the discussion and clinical implications of the study.

Response: The resource burden associated with self-harm is of relevance to the clinical services responsible for the mental health care of prisoner populations. The care planning process used to monitor those at increased risk of self-harming behaviour, the Assessment, Care in Custody and Teamwork process (ACCT), has significant staff resource implications for both prison and clinical services. We agree that this point is better placed in the Discussion: Clinical implications sub-section: it is now moved and reworded as the second sentence in this section of the paper as follows: ‘Any reduction in self-harming behaviour occasioned by the suggested environmental changes (e.g., reduced use of single cell placements) has the potential to improve prisoner well-being and to reduce the use of resource intensive care planning processes such as the Assessment, Care in Custody and Teamwork (ACCT) process employed in prisons in England and Wales.’

4

The last sentence of the second paragraph requires improvements in language and clarity, along with more precise reporting of the odds ratio, including a confidence interval.

Response: We thank the Reviewer for noting this. No confidence interval was reported for this variable. We have reflected on the difficulty the reader might have when interpreting this data, despite the addition of other data reported in the study such as the Χ2 value and p value. We have opted to remove quoting of any OR value. Therefore, we have rephrased this to read “One case-control study additionally identified that frequent cell moves within prison over the previous two years was associated with self-harm behaviour5.”

5

In the third paragraph, the authors discuss a wide array of prison environmental factors that might contribute to self-harm, but the direction of these associations should be clarified. Furthermore, while the authors note the current uncertainty around the strength of these effects, it is important to recognize that the strength of associations varies significantly depending on the context and prison population. This aspect warrants some elaboration.

Response: We have clarified the direction of each of the relationships implicated in the wider literature. Whilst we fully acknowledge the Reviewer’s point that the strength of associations is likely to vary depending on context and prison population, we have reserved our acknowledgement of this point for the discussion as we think it is of greater relevance when discussing the value of further replication studies than in introducing this single-site study. Thus, it is addressed in the fourth sentence of the Discussion: Future research sub-section as follows: ‘Replication of the study findings in research in prisons of varying security, location and function, and in the female prison population, will improve generalisability and allow for cross-comparison of the strength of effects.’

6

The introduction incorporates multiple references to specifically female samples, which is not inherently problematic. However, given that this study focuses on an all-male sample, it is important to clearly highlight when employing such references.

Response: We thank the Reviewer for this observation and have acknowledged this by amending the third paragraph of the introduction as follows: “In addition, most existing studies examine female prisoner populations. Longitudinal study designs could provide high-quality evidence regarding these relationships in both male and female populations, which could in turn inform the clinical assessment and management of self-harm in this population and suicide prevention strategies in prisons.”

7

The aims appear to be structured as a primary and a secondary aim, suggesting a hierarchy of importance. If my understanding is correct, this distinction should be explicitly stated.

Response: As suggested by the Reviewer, we have removed to secondary aim. Please see response to Intro point 8 below.

8

The introduction provides minimal context regarding the secondary aim. This aim might be considered for removal. Alternatively, it would be beneficial to elaborate on the design of a wider array of previous studies (including sample size, and statistical approaches), and the insights they have offered for the purpose of this aim.

Response: As suggested by the Reviewer, we have removed the secondary aim.

9

Mentioning that the article was part of a wider project is redundant for the aims paragraph. However, as a reader, I would appreciate a summary description of the SHAPE project (if this is what the authors are referring to here) in the methods section, but I leave this entirely up to the authors to decide.

Response: We have moved this and provided a summary description of the SHAPE project in the Procedures and measures sub-section of the Methods section.

Methods

1

Please describe the attrition, for example, how many participants were excluded due to their English language skills or decision-making capacity, respectively.

Response: We are grateful to the Reviewer for requesting further detail here. We have amended the text from the fourth sentence of the second paragraph onwards to separate those not included (n=147) and those excluded on records-based and in-person screening and provide details of the breakdown in each group. We have made the corresponding edits to figure 1.

2

Participants lost due to suicide during the follow-up period should be mentioned, even if there were no cases, if this was measured.

Response: We have clarified that no participants died by suicide during follow-up by adding a sentence at the end of the second paragraph of the Results – Cohort characteristics sub-section.

3

Category B prisons are contrasted with Categories C and D but should also be contrasted with Category A.

Response: We have added the phrase “and a lesser emphasis on security than those in Category A” to the second sentence of the Methods: Sample sub-section.

4

A strength of this study is that the sample corresponds to the general prison population on key factors. However, note observed differences between prisoners awaiting trial, those sentenced, and those detained under immigration law in the final sample.

Response: We have amended the first paragraph of the ‘Results – Cohort characteristics’ sub-section to reflect the change in proportion of those on remand and sentenced in the final cohort.

5

It is unclear whether the sample comprises “all male prisoners who had recently arrived” or an all-male sample.

Response: We thank the Reviewer for highlighting this potential source of confusion. It is stated in the first paragraph of the Methods section that the study site is a men’s prison. Thus, we have rephrased the first sentence of the second paragraph to remove any further references to sex, to avoid confusion. It now reads “The study population consisted of all prisoners arriving into the study prison between 17 March and 16 July 2022.”

6

Briefly describe the simple random sampling approach used in this study.

Response: Our sampling approach involved a series of weekly random samples during 18 weeks of recruitment between March to July 2022. We have amended the text in the third sentence of the second paragraph of Methods: Sample sub-section to clarify this.

7

If I understand Figure 1, the sentence describing eligibility criteria in the sample segment is need of correction. Should it state “aged 18 or older, male, and current prison spell had lasted less than one month”?

Response: We thank the Reviewer for spotting this error in classification of the length of current prison spell as an exclusion criterion and have corrected it to classify ‘having a current prison spell lasting less than one month’ as an inclusion criterion. We have made the corresponding edits to figure 1 and adjusted the numbers included/not included and excluded/not-excluded accordingly.

8

The average length of follow-up was 73.5 days. Please indicate the standard deviation or other measure of variability.

Response: We have added the standard deviation, 24.1 days, to the manuscript.

9

The instruments used such as DUDIT and CTQ, are insufficiently described and not referenced. Furthermore, it is not clear how information from these instruments were utilized in this study and if they were administered alongside the supplemented questionnaires.

Response: We are grateful to the Reviewer for the opportunity to clarify this point. Although these instruments were administered alongside the study questionnaire, data collected using them were not used in the present study. Amongst the data used in this study, data for one confounder factor (violent offence history) and one exposure factor (vape use) were assessed in the baseline interview. The remaining data collected in the baseline interview were for use in a separate study within the SHAPE project which is in preparation. We have therefore amended the text to reflect this and have moved the first two paragraphs of the sub-section ‘Procedures and measures’ to be below the summary description of the SHAPE project (see response to Intro 9). We have renamed this sub-section ‘Measures and procedures’ to reflect the order in which these are now described in the revised paragraph.

10

In the methods section, it is mentioned that this was an exploratory pilot study, a description with which I agree. However, the language in the abstract, aims, and discussion needs revision to consistently reflect this.

Response: We have adjusted the descriptions of the study throughout the manuscript to consistently reflect this.

11

The three participants who withdrew their consent were not accounted for in the comparison of non-identifiable demographic data, as they seem to be missing from the stated n=126, referencing the last paragraph in the sample segment. This figure should likely be adjusted to 129 and comparative test rerun accordingly.

Response: We have adjusted this figure to n=129 and rerun the comparative tests.

12

Throughout the document, ensure that statistical notation is used correctly, such as changing ‘mean’ to italicized M, and consistently reporting odds ratios with or without the 95% numeral in accordance to applied reporting standards.

Response: We thank the Reviewer for this observation. We have amended the manuscript throughout to include statistical notation where appropriate (i.e. in Tables and when reporting numerical results with a mathematical operator). However, we have retained the use of terms rather than symbols in the narrative text to improve readability. This is consistent with APA style guidelines. Nevertheless, we are very happy to defer to the Editor for the final decision here. We have addressed the reason for not presenting the confidence intervals for the OR quoted in the introduction (now removed; see Intro point 4). We consistently report 95% CI for other odds ratios throughout the manuscript.

13

The independence of the researchers from the regular prison staff should be clarified in the methods section. If there was no independence, this raises questions for the limitations section, such as the potential impact on informed consent and the data collected. Additionally, the lack of researcher blindness to participants could have influenced the findings. The potential effects of this potential lack of blinding and independence on the study’s reliability and impartiality need to be addressed.

Response: We have clarified that researchers were independent from the regular prison staff (final paragraph of the Methods: Measures and procedures sub-section). In the penultimate paragraph of this sub-section, we have clarified that TS was blinded to all participants’ exposure values except for vape Use at the time of carrying out the exit interviews.

14

SHAPE is used as an abbreviation but not explained.

Response: This is now explained as per response to Intro point 9.

15

The decision to include vape use as an exposure for self-harm warrants further explanation. Since this variable has been singled out based on the authors’ unpublished results, it is crucial to offer a rationale for its inclusion. Additionally, clarifying why vape use is considered an environmental risk factor, rather than a risk factor related to individual behaviour, is needed.

Response: We have now clarified that this variable was included, alongside Covid-19 in-cell isolation, based on the unpublished results of the parallel qualitative study, by including it in the summary of the preliminary results of this study (see response to Method

---

## [Decision Letter · Decision Letter 1]

28 Aug 2024

PONE-D-24-04206R1Environmental risk factors for self-harm during imprisonment: a prospective cohort studyPLOS ONE

Dear Dr. Stephenson,

Thank you for submitting your manuscript to PLOS ONE. The manuscript reviewer recorded some minor concerns about regressive analyses, as well as a need to provide more context for these analyses. As such, the reviewer recommended further revisions be made.

In my judgment, these revisions are minor, and therefore I would be pleased to offer a minor revision decision. Still, since the timescale is shorter than can be accommodated for the swift turnaround required for publication, please let me know if you would prefer an extended deadline.

On a final note, I suggest that you ignore the advice on APA referencing style, since the journal follows the Vancouver referencing style.

We look forward to receiving your revised manuscript.

Kind regards,

Nasrul Ismail

Academic Editor

PLOS ONE

Journal Requirements:

Reviewers' comments:

Reviewer's Responses to Questions

**Comments to the Author**

1. If the authors have adequately addressed your comments raised in a previous round of review and you feel that this manuscript is now acceptable for publication, you may indicate that here to bypass the “Comments to the Author” section, enter your conflict of interest statement in the “Confidential to Editor” section, and submit your "Accept" recommendation.

Reviewer #1: All comments have been addressed

2. Is the manuscript technically sound, and do the data support the conclusions?

Reviewer #1: Partly

3. Has the statistical analysis been performed appropriately and rigorously? 

Reviewer #1: Yes

4. Have the authors made all data underlying the findings in their manuscript fully available?

Reviewer #1: No

5. Is the manuscript presented in an intelligible fashion and written in standard English?

Reviewer #1: Yes

6. Review Comments to the Author

Reviewer #1: General comments

I would like to express my gratitude to the editor for the opportunity to review this revised article and to the authors for the significant improvements they have made. The authors have effectively addressed most of my previous comments and have provided well-reasoned explanations for the areas where they chose not to make revisions. However, I still hold one major concern (see R1.17) regarding the analytic approach and a few instances where my queries remain unsatisfactorily addressed. I have detailed these below (for reference, "R1.3" pertains to the third specific comment in my previous review, while "R2.1" refers to new specific comments in this review).

Generally, I would like to invite the authors to revise the study to better reflect the APA guidelines (or the chosen guideline to be followed consistently). For example and based on my understanding, “(17/218)” should be written as “17 out of 218” (or simply as a percentage). Additionally, the number of decimal places varies across the manuscript, and the numerical representation of thousands should be consistent (i.e., 1,000 or 1000). Tables should be capitalized in text and numbers that never exceed one should not be written with a zero (i.e., p = .03 is correct, while p = 0.03 is incorrect according to APA). I also noticed a missing “p=” in the first paragraph of the regression analyses. Furthermore, the text could benefit from minor edits to improve clarity and reduce ambiguity. The tables, in particular, require such editing. For example, the sentence regarding the exclusion of participants in the sample section needs revision, and there is a repetition of “due to [with subsequent information]” in the same section.

PLOSE ONE’s criteria for publication

1. This article contributes to the base of academic knowledge and presents results of original research by exploring prison environmental factors for self-harm and estimating their effects.

2. I have found no evidence of these findings being reported elsewhere, except in a preprint format, which adheres to PLOS ONE’s criteria for publication: https://www.medrxiv.org/content/10.1101/2024.01.31.24302059v1.

3. The statistical analyses were performed to a good technical standard but improvements are recommended, as elaborated on below.

4. Conclusions are presented in an appropriate fashion but with remaining concerns regarding the statistical approach and interpretations, revisions are recommended.

5. The article is presented in an intelligible fashion and is written in standard English.

6. The research appears to adhere to the ethical standards and it received approval from the relevant ethical board as well as the national prison and probation services.

7. The article generally follows appropriate reporting standards but requires some improvements. The data have not been made available to others, a decision that is sound given the design and participants of the study. Data requests are directed to relevant services.

Introduction

Comment from the original submission

R1.3.

If available, use an academic or other reference for this intervention.

New comment

R2.1

The fourth sentence in the second paragraph of the introduction, starting with “[t]he authors…”, seems to use reference (4) but this is not clear.

Methods

Comment from the original submission

R1.17

To my understanding, the study collected data on certain risk factors at baseline, but with most risk factor information gathered during the prospective follow-up period. Further, the data for the outcome of self-harming behaviour was also collected during this follow-up period. Logistic regression analyses were conducted, without considering follow-up time or time-to-event in the statistical analyses.

In a prospective cohort study, it is possible to analyse risk factors collected during the same follow-up period as the outcome. However, this approach has notable limitations, such as the risk of reverse causality, as the authors are aware. Understanding cause and effect – or as in here, when estimating effect sizes – requires proper time orientation, i.e., an exposure (risk factor) must occur before the outcome. This also requires having a sufficient level of detail in the data regarding when exposure and outcome occur so that the temporal order can be accurately distinguished (or explored/estimated). Furthermore, it requires an adequate model of analysis. The authors briefly discuss reverse causality as a limitation, but I find this to be insufficient. I recommend that the authors improve their analytic approach pertaining to the regression analyses approach. This could involve using alternative methods or multiple methods that consider the varying characteristics of risk factors, and reanalyse their data. Favourably, with time-to-event and, if appropriate, subsequent censoring taken into account. If my understanding is correct, approximate time-sensitive data is available to the authors through prison medical records. This would significantly improve the article. If the authors decide to revise their analytic approach, I further recommend that the authors reconsider the number of risk factors included in the analysis of the outcome. Given the low number of participants and events, the current number of risk factors may be too extensive, even for an exploratory pilot study.

These revisions are important partly due to the authors’ allusive recommendations on prison services based on interpretations of the data that are currently weakly supported by the data due to mentioned limitations. Furthermore, the authors state and acknowledge that a self-harm event increases the probability of encountering certain risk factors after the event, making the timing of data collection even more crucial when employing risk factors in statistical analyses collected before and after potential events.

New comment

R2.1

In the first paragraph of the sample section, the authors state that previous longitudinal research at the same study site has documented a higher than average prevalence of self-harm, suggesting it is a suitable study location. However, this suitability depends on the context. While it may offer valuable insights, it could also potentially lower the generalizability of the findings, depending on the study’s aims. This increased prevalence might be influenced by systematic biases, such as the characteristics of the sample or the prison environment. Please consider this and revise accordingly.

Additionally, as a reader, I am interested in the previous research mentioned. If the authors deem it appropriate, I would recommend including a reference to this prior research.

Figures and tables

New comment

R2.1

The tables, especially Table 2 and 3 needs improvement, focusing on removing redundant information, minimizing double reporting in text and tables, as well as general readability.

Results, discussion and conclusions

Comments from the original submission

R1.1

The information of discordance is important. However, the paragraph with this information could do with streamlining.

R1.3

Thank you for a good response. However, here I believe that the authors have misunderstood parts of my comment, in part based on my comment’s ambiguity. My stated concern was primarily regarding the methodology used regarding the prison wing that was closed at the start of the study (i.e.: One prison wing was closed at the start of the study but gradually reopened during the second half of the study and reached approximately 50% capacity at end of the follow-up period. The total occupancy for this particular wing was imputed as 25% of capacity throughout the period of reopening based on consultation with prison officials). Please reconsider this comment and my apologies if I am misunderstanding the given response.

New comments

R2.1

As the aim was to “…estimate the effect sizes…”, the discussion lacks discussion regarding the effect sizes in the current study. Please revise the discussion to include this, or alternatively, adjust the aim if this better align with the study’s underlying purpose.

R2.2

The discussion appropriately highlights the novelty of several results, which is commendable. However, it appears to lack adequate contextualization of these findings within the specific setting of the study, such as the remand context, the prison environment with a higher-than-average prevalence of self-harm, the male sample, and the location in England. Providing a more detailed contextualization of the results could enhance their relevance and usefulness, after considering comment R1.17.

R2.3

In the clinical implications section, it is (kind of) mentioned that the authors suggest a reduced use of single-cell placements. This either needs to be explicitly stated as a recommendation from the authors or revised. My interpretation of the results, in combination with the discussion above this section, is that the results are preliminary and in need of replication before making such recommendations, partly as the findings may vary across different subgroups and contexts. Therefore, such a recommendation may currently be most valuable to the specific prison in question and for guiding future studies at this time. Furthermore, this revision should also reflect the R1.17-comment. Nevertheless, I leave the final interpretation and wording up to the authors.

7. PLOS authors have the option to publish the peer review history of their article (what does this mean? ). If published, this will include your full peer review and any attached files.

**Do you want your identity to be public for this peer review?** For information about this choice, including consent withdrawal, please see our Privacy Policy .

Reviewer #1: No

---

## [Author Response · Author response to Decision Letter 2]

21 Nov 2024

Dear Dr Ismail,

Thank you for the opportunity to revise our manuscript further in relation to the comments in the first reviewer’s second review. Alongside our submission of a manuscript with further revisions, in this letter we have outlined our response to each point raised in the reviewer’s second review. We have used the reviewer’s numbering of comments (1.3, 2.1 etc) but added a letter indicating the section of the paper (e.g. A1.3 for Abstract and introduction) to avoid confusion.

General comments

Reviewer’s comment: I would like to express my gratitude to the editor for the opportunity to review this revised article and to the authors for the significant improvements they have made. The authors have effectively addressed most of my previous comments and have provided well-reasoned explanations for the areas where they chose not to make revisions. However, I still hold one major concern (see R1.17) regarding the analytic approach and a few instances where my queries remain unsatisfactorily addressed. I have detailed these below (for reference, "R1.3" pertains to the third specific comment in my previous review, while "R2.1" refers to new specific comments in this review).

Generally, I would like to invite the authors to revise the study to better reflect the APA guidelines (or the chosen guideline to be followed consistently). For example and based on my understanding, “(17/218)” should be written as “17 out of 218” (or simply as a percentage). Additionally, the number of decimal places varies across the manuscript, and the numerical representation of thousands should be consistent (i.e., 1,000 or 1000). Tables should be capitalized in text and numbers that never exceed one should not be written with a zero (i.e., p = .03 is correct, while p = 0.03 is incorrect according to APA). I also noticed a missing “p=” in the first paragraph of the regression analyses. Furthermore, the text could benefit from minor edits to improve clarity and reduce ambiguity. The tables, in particular, require such editing. For example, the sentence regarding the exclusion of participants in the sample section needs revision, and there is a repetition of “due to [with subsequent information]” in the same section.

Response: We are grateful to the reviewer for highlighting discrepancies in the manuscript, and have now corrected these throughout. Regarding the presentation of numbers, we have applied the PLOS statistical reporting guidelines throughout the manuscript. Our responses to the numbered points raised are included in the sections below.

Abstract and introduction

Comment A1.3

Reviewer’s comment in first review: The final sentence of the first paragraph, which describes self-harm as a resource burden for service providers, lacks citations. Moreover, its relevance to the paper may be questioned. If the authors deem this information important to the introduction, further elaboration and supporting references are needed, but may also be saved for the discussion and clinical implications of the study.

Response: The resource burden associated with self-harm is of relevance to the clinical services responsible for the mental health care of prisoner populations. The care planning process used to monitor those at increased risk of self-harming behaviour, the Assessment, Care in Custody and Teamwork process (ACCT), has significant staff resource implications for both prison and clinical services. We agree that this point is better placed in the Discussion: Clinical implications sub-section: it is now moved and reworded as the second sentence in this section of the paper as follows: ‘Any reduction in self-harming behaviour occasioned by the suggested environmental changes (e.g., reduced use of single cell placements) has the potential to improve prisoner well-being and to reduce the use of resource intensive care planning processes such as the Assessment, Care in Custody and Teamwork (ACCT) process employed in prisons in England and Wales.’

Reviewer’s comment in second review: If available, use an academic or other reference for this intervention.

Response: We have not identified any academic reference for the ACCT process, however we have added references to two publicly available documents on the UK Ministry of Justice website which outline this process.

Ministry of Justice. Prison Service Instruction 64/2011: Management of Prisoners at Risk of Harm to Self, to Others and from Others (Safer Custody). London: Ministry of Justice; 2011. [Available online at https://www.gov.uk/government/publications/managing-prisoner-safety-in-custody-psi-642011, accessed on 21/10/24.]

Ministry of Justice. Annex to PSI 64/2011 (ACCT). London: Ministry of Justice; 2020. [Available online at https://www.gov.uk/government/publications/managing-prisoner-safety-in-custody-psi-642011, accessed on 21/10/24].

See response to comment R2.3 regarding changes the content of the text where these references are cited.

Comment A2.1

Reviewer’s comment in second review: The fourth sentence in the second paragraph of the introduction, starting with “[t]he authors…”, seems to use reference (4) but this is not clear.

Response: We are grateful to the reviewer for spotting this ambiguity and have updated the wording of that sentence to read “The same systematic review and meta-analysis also identified…”.

Methods

Comment M1.17

Reviewer’s comment in first review: I would like confirmation that risk factors were assessed at baseline and not during the three-month follow-up period, as assumed from the prospective design, but this is not entirely clear.

Response: Exposures were assessed during follow-up as outlined in paragraphs 1 and 2 of the Methods: Measures and procedures sub-section except for vape use which was assessed during the baseline interview (see response to Methods point 9). We have amended the wording of paragraph 2 of this sub-section so that sentences 4 and 5 now read “Vape use was assessed by self-report at baseline interview. All other exposures were measured using routinely collected data on participants’ NOMIS prison record during follow-up.”

Reviewer’s comment in second review: To my understanding, the study collected data on certain risk factors at baseline, but with most risk factor information gathered during the prospective follow-up period. Further, the data for the outcome of self-harming behaviour was also collected during this follow-up period. Logistic regression analyses were conducted, without considering follow-up time or time-to-event in the statistical analyses.

In a prospective cohort study, it is possible to analyse risk factors collected during the same follow-up period as the outcome. However, this approach has notable limitations, such as the risk of reverse causality, as the authors are aware. Understanding cause and effect – or as in here, when estimating effect sizes – requires proper time orientation, i.e., an exposure (risk factor) must occur before the outcome. This also requires having a sufficient level of detail in the data regarding when exposure and outcome occur so that the temporal order can be accurately distinguished (or explored/estimated). Furthermore, it requires an adequate model of analysis. The authors briefly discuss reverse causality as a limitation, but I find this to be insufficient. I recommend that the authors improve their analytic approach pertaining to the regression analyses approach. This could involve using alternative methods or multiple methods that consider the varying characteristics of risk factors, and reanalyse their data. Favourably, with time-to-event and, if appropriate, subsequent censoring taken into account. If my understanding is correct, approximate time-sensitive data is available to the authors through prison medical records. This would significantly improve the article. If the authors decide to revise their analytic approach, I further recommend that the authors reconsider the number of risk factors included in the analysis of the outcome. Given the low number of participants and events, the current number of risk factors may be too extensive, even for an exploratory pilot study.

These revisions are important partly due to the authors’ allusive recommendations on prison services based on interpretations of the data that are currently weakly supported by the data due to mentioned limitations. Furthermore, the authors state and acknowledge that a self-harm event increases the probability of encountering certain risk factors after the event, making the timing of data collection even more crucial when employing risk factors in statistical analyses collected before and after potential events.

Response: Due to limited time resources, we did not plan to collect time-sensitive data for exposure factors in this pilot and so planned logistic regression analyses between exposure factors and binary outcomes instead. Unfortunately, alternative methods involving collection of time-sensitive data are now no longer available to us, due to loss of access (via NOMIS custodial records) to the exposure data for participants after they move on from the study prison. We are grateful to the reviewer for emphasising the importance of the limitation this analysis method places on any conclusions about causality in the relationship between exposure factors and outcomes.

We have thus added the following sentence to the second paragraph of the Methods: Measures and procedures sub-section to signpost the lack of time-to-event data and justify the choice of methods, and hope the reviewer agrees that this provides sufficient signposting that the subsequent acknowledgement of it as a limitation in the Discussion section is sufficient:

“Due to limited resources, the timing of exposures and outcomes was not assessed in this study.”

We have also edited the fourth sentence of the Discussion: Future research sub-section to read “Future studies assessing time-to-event data could help to discern the direction of any effects underlying the associations observed” with the intention of further emphasising the above.

To emphasise the limited scope of this pilot study, we have amended the manuscript title to include the term “pilot study” and have added to the study’s aims an estimation of the attrition rate in the study population to inform future research.

We have removed the recommendations for prison services in the clinical implications sub-section (see response to comment R2.3) in further acknowledgement that this is pilot data which requires replication.

For the reviewer’s information, we are now funded by NIHR to conduct a larger prospective cohort study (with sample size calculations based on strength of associations observed in this pilot) where we plan to analyse time-to-event data for a smaller set of exposure variables using survival analysis approaches. We are grateful to the reviewer for his/her helpful thoughts in this area which helped to strengthen our application and analytic approach.

Comment M2.1

Reviewer’s comment: In the first paragraph of the sample section, the authors state that previous longitudinal research at the same study site has documented a higher than average prevalence of self-harm, suggesting it is a suitable study location. However, this suitability depends on the context. While it may offer valuable insights, it could also potentially lower the generalizability of the findings, depending on the study’s aims. This increased prevalence might be influenced by systematic biases, such as the characteristics of the sample or the prison environment. Please consider this and revise accordingly.

Additionally, as a reader, I am interested in the previous research mentioned. If the authors deem it appropriate, I would recommend including a reference to this prior research.

Response: We agree with the reviewer that the suitability of a study location depends on several contextual factors and have clarified in that paragraph that we refer to suitability “in terms of feasibility for a pilot study”.

We have acknowledged the potential impact of systematic biases referred to in the reviewer’s comment by adding the following to the third paragraph of the Discussion: Limitations sub-section:

“The study was conducted in a prison with a higher-than-average prevalence of self-harm, and this higher prevalence may be influenced by unmeasured systematic biases such as the characteristics of the prison environment. Such biases would lower the generalisability of findings to other prison settings and sub-populations.”

The prior research informing our decision about study location is now referenced in the manuscript (reference 14).

Figures and tables

Comment F2.1

Reviewer’s comment: The tables, especially Table 2 and 3 needs improvement, focusing on removing redundant information, minimizing double reporting in text and tables, as well as general readability.

Response: We thank the reviewer for highlighting issues with the manuscript tables. We have made several changes across tables 2, 3 and 4 to address the points raised:

• Removed redundant text in column labels

• Harmonised terminology across tables and manuscript e.g. self-harm episode corrected to self-harm behaviour

• Removed dual unit labels, separating parameters into new rows where necessary (e.g. SD and IQR in table 3)

• Removed redundant information, e.g. Missing data rows where no data missing for that variable

• Rearranged variables for readability, e.g. Prisoner-staff ratio and Emergency bell response rate moved below Prisoner-Listener ratio in table 3 because we report M and SD for these variables rather than Mdn and IQR as for variables above

• Formatted of numbers brought into line with changes made across the manuscript to present max 2dps, e.g. p values in table 4 reformatted.

Results, discussion, and conclusions

Comment R1.1

Reviewer’s comment in first review: Key information in regards to the validity of the secondary aim is the correlation or concordance or NSSI and NSSI ideation and could be reported. Auxiliary, the comparison between the 14/82 registered versus the 13/78 self-reported events of NSSI suggests a need to assess their concordance or discordance as suggested by the authors in the methods section. This also have bearing on the third limitation.

Response: We have added text to the second paragraph of the ‘Results – Cohort characteristics’ sub-section on the discordance between the records-based and self-reported self-harm behaviour events (6.1% of cases), and on the concordance between records-based self-harm behaviour and self-reported self-harm ideation. We are grateful to the Reviewer for highlighting this and the implications for further research. We have amended the final sentence of the Discussion: Future research sub-section to read “Discordance between outcomes assessed via prison medical records and self-report was low (6.1%) but may nonetheless have implications for future study designs in this area.”

Reviewer’s comment in second review: The information of discordance is important. However, the paragraph with this information could do with streamlining.

Response: We thank the author for highlighting that this paragraph was in need of streamlining. We have now edited it down for clarity so that it now reads:

“The event rate for any self-harm behaviour during follow-up was 17.1% (14/82) based on medical records and 16.7% based on self-report (13/78), with discordance between the two measures in 5 of 78 (6.1%) cases for whom data was available such that 19.5% of participants (16/82) had a positive classification for self-harm behaviour on at least one measure. The event rate for any self-reported self-harm ideation was 28.2% (22/78). The concordance between self-harm behaviour outcome from records and self-harm ideation outcome from self-report was 82.1% (64/78) amongst those for whom data was available for both measures. No participants died from suicide during follow-up.”

Comment R1.3

Reviewer’s comment in first review: I am concerned about the methodology used by the authors in treating the staff-to-prisoner ratio variable, specifically the imputation of numbers based on prison conditions in a new ward. This may have influenced the negative association found for prisoners per staff member

---

## [Decision Letter · Decision Letter 2]

12 Jan 2025

Environmental risk factors for self-harm during imprisonment: a pilot prospective cohort study

PONE-D-24-04206R2

Dear Dr. Stephenson,

We’re pleased to inform you that your manuscript has been judged scientifically suitable for publication and will be formally accepted for publication once it meets all outstanding technical requirements.

Kind regards,

Dr Nasrul Ismail

Academic Editor

PLOS ONE

Additional Editor Comments (optional): N/A

Reviewers' comments: Accept

Reviewer's Responses to Questions 

**Comments to the Author **

1. If the authors have adequately addressed your comments raised in a previous round of review and you feel that this manuscript is now acceptable for publication, you may indicate that here to bypass the “Comments to the Author” section, enter your conflict of interest statement in the “Confidential to Editor” section, and submit your "Accept" recommendation.

Reviewer #2: Accept

2. Is the manuscript technically sound, and do the data support the conclusions?

Reviewer #2: Yes

3. Has the statistical analysis been performed appropriately and rigorously? 

Reviewer #2: Yes

4. Have the authors made all data underlying the findings in their manuscript fully available?

Reviewer #2: No

5. Is the manuscript presented in an intelligible fashion and written in standard English?

Reviewer #2: Yes

6. Review Comments to the Author

Reviewer #2: (No Response)

7. PLOS authors have the option to publish the peer review history of their article (what does this mean? ). If published, this will include your full peer review and any attached files.

**Do you want your identity to be public for this peer review?** For information about this choice, including consent withdrawal, please see our Privacy Policy .

Reviewer #2: No

---

## [Editor Report · Acceptance letter]

PONE-D-24-04206R2

PLOS ONE

Dear Dr. Stephenson,

I'm pleased to inform you that your manuscript has been deemed suitable for publication in PLOS ONE. Congratulations! Your manuscript is now being handed over to our production team.

Kind regards,

on behalf of

Dr. Nasrul Ismail

Academic Editor

PLOS ONE